# END-TO-END (INSTANCE)-IMAGE GOAL NAVIGATION THROUGH CORRESPONDENCE AS AN EMERGENT PHENOMENON

**Guillaume Bono, Leonid Antsfeld, Boris Chidlovskii, Philippe Weinzaepfel & Christian Wolf**
Naver Labs Europe Meylan, France
{firstname.lastname}@naverlabs.com

## ABSTRACT

Most recent work in goal oriented visual navigation resorts to large-scale machine learning in simulated environments. The main challenge lies in learning compact representations generalizable to unseen environments and in learning high-capacity perception modules capable of reasoning on high-dimensional input. The latter is particularly difficult when the goal is not given as a category ("*ObjectNav*") but as an exemplar image ("*ImageNav*"), as the perception module needs to learn a comparison strategy requiring to solve an underlying visual correspondence problem. This has been shown to be difficult from reward alone or with standard auxiliary tasks. We address this problem through a sequence of two pretext tasks, which serve as a prior for what we argue is one of the main bottleneck in perception, extremely wide-baseline relative pose estimation and visibility prediction in complex scenes. The first pretext task, cross-view completion is a proxy for the underlying visual correspondence problem, while the second task addresses goal detection and finding directly. We propose a new dual encoder with a large-capacity binocular ViT model and show that correspondence solutions naturally emerge from the training signals. Experiments show significant improvements and SOTA performance on the two benchmarks, *ImageNav* and the *Instance-ImageNav* variant, where camera intrinsics and height differ between observation and goal.

## 1 INTRODUCTION

Goal oriented visual navigation is usually addressed through large-scale training in simulation, followed by sim2real transfer. While decision taking has not yet been solved either, recent research provides evidence that perception is a major bottleneck with several challenges: learning representations required for planning; extracting 3D information, difficult when depth is not available or not reliable; and, generalizing to unseen environments, which is challenging given the limited number of existing training environments.

The perception module of an agent needs to address several skills, which include detecting navigable space and obstacles, detecting exits necessary for long horizon planning, detecting goals and estimating the agent's relative pose with respect to them, see Figure 1. The detection of visual goals given by exemplars requires to solve a partial matching task, which in essence is a *wide-baseline visual correspondence problem*. They are classical in computer vision and at heart of methods in visual localization and relative pose estimation (Humenberger et al., 2022; Revaud et al., 2019; Sarlin et al., 2020). We argue that in navigation, however, they did not get the attention they deserve.

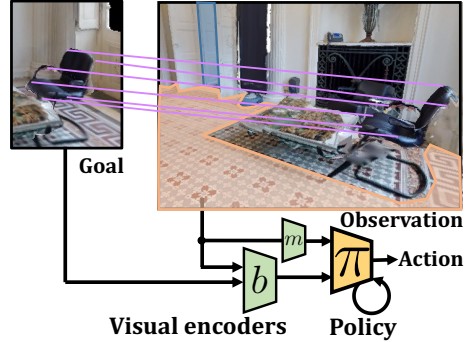

Figure 1: Navigation skills include detecting navigable space, exits, and the agent's relative pose wrt. the goal. The correspondence solutions required by pose emerge from training with pretext tasks.

Instead, robot perception is addressed through scene reconstruction, for instance with SLAM (Chaplot et al., 2020b; Lluvia et al., 2021; Thrun et al., 2005) or by putting the full burden of perception on a visual encoder trained end-to-end from objectives like RL (Jaderberg et al., 2017; Zhu et al., 2017) or imitation learning (Ding et al., 2019). The former does not address goal detection, which needs to be outsourced to an external component. The latter, when trained on tasks like *ImageNav*, attempts to solve the problem implicitly without direct supervision through weak learning signals.

This has been shown to be difficult, witnessed by the wide usage of more complex sensors for the *ImageNav* task compared to tasks like *ObjectNav*, where the goal is specified through its category. For several years, state-of-the art methods for *ImageNav* used panoramic images consisting of 4 observed images taken at angles of 90° (see Figure 2), which facilitates learning the underlying pose estimation task from weak learning signals, but is restrictive in terms of robotic applications. Only recently the field switched to mono-view input.

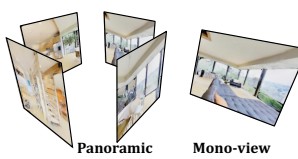

Figure 2: Panoramic vs. mono-view input.

We propose a new method for goal oriented visual navigation, which introduces a sequence of pretext tasks of directional learning and visual correspondence. We take advantage of recent advances in unsupervised model pre-training for low-level scene understanding and build on the work of Weinzaepfel et al. (2022), which proposes a type of multi-view pretext task, namely *cross-view completion*, a 3D variant of masked image modeling. We show that the underlying correspondence problem solved by this model is particularly relevant to the *ImageNav* problem and adapt this model through a new dual-encoder model.

In recent work (Krantz et al., 2023), the same problem is addressed with explicit feature matching combined with a modular map-based method. In contrast, our method does not rely on explicit correspondence calculations, the entire agent is differentiable and the perception module is comprised of a combination of a monocular and a binocular encoder. We do, however, show that correspondence solutions emerge from pre-training through the cross-attention behavior, see Figure 4 in the experimental section.

We present the following contributions: (i) We define a pretext task of extremely wide-baseline relative pose estimation highly correlated with navigation and positioning cues from visual input. We also introduce a new dataset tailored for navigation; (ii) We couple relative pose estimation with the estimation of visibility, which we link to the specific capacity to decide whether to explore or to exploit; (iii) We additionally perform self-supervised pre-training for cross-view completion ("*CroCo*") (Weinzaepfel et al., 2022) and show its impact; (iv) We show that correspondence solutions emerge from pre-training with these tasks. (v) We propose a dual visual-encoder architecture based on vision transformers and cross-attention, which we integrate into an end-to-end agent and, as proof-of-concept, into a modular architecture; (vi) We obtain SOTA performance on two standard benchmarks, *ImageNav* and *Instance-ImageNav*.

## 2 RELATED WORK

**Visual navigation** — navigation has been classically solved in robotics using mapping and planning (Burgard et al., 1998; Macenski et al., 2020; Marder-Eppstein et al., 2010), which requires solutions for mapping and localization (Bresson et al., 2017; Labbé & Michaud, 2019; Thrun et al., 2005), for planning (Konolige, 2000; Sethian, 1996) and for low-level control (Fox et al., 1997; Rösmann et al., 2015). These methods depend on accurate sensor models, filtering, dynamical models and optimization. End-to-end trained models directly map input to actions and are typically trained with RL (Jaderberg et al., 2017; Mirowski et al., 2017; Zhu et al., 2017; Bono et al., 2024) or imitation learning (Ding et al., 2019). They learn representations, either flat recurrent states or occupancy maps (Chaplot et al., 2020b), semantic maps (Chaplot et al., 2020a), latent metric maps (Beeching et al., 2020b; Henriques & Vedaldi, 2018; Parisotto & Salakhutdinov, 2018); topological maps (Beeching et al., 2020a; Chaplot et al., 2020c; Shah & Levine, 2022), self-attention (Chen et al., 2022a; Du et al., 2021; Fang et al., 2019; Reed et al., 2022) or implicit representations (Marza et al., 2023). Our method is end-to-end trained but adds pretext tasks for perception.

**Goal-oriented navigation** — In the easier *ObjectNav* setting, the goal is provided as a category and a detector can encode object shapes in model parameters, trained explicitly for detection, e.g. with semantic maps (Chaplot et al., 2020a), map-less object detectors (Savva et al., 2019) or image segmenters (Maksymets et al., 2021), or end-to-end through the navigation loss. *ImageNav* provides the goal as an exemplar image and is a significantly harder task, requiring the perception model to learn a matching strategy itself. Most work are based on end-to-end training (Zhu et al., 2017; Mezghani et al., 2022; Majumdar et al., 2023), potentially supported through self-supervised losses (Majumdar et al., 2022b). Modular approaches have also been proposed (Das et al., 2018a; Wu et al., 2022). The very recently introduced *Instance-ImageNav* task requires to handle different camera intrinsics and heights between observation and goal, which prior work does with explicit feature matching (Krantz et al., 2023). We make it possible to address the *ImageNav* and *Instance-ImageNav* with end-to-end methods in the challenging mono-view setting through new pretext tasks.

**Pretext tasks in CV and navigation** — widely used in NLP and CV, pretext tasks aim at learning representations followed by fine-tuning for particular tasks (Devlin et al., 2019; Yuan et al., 2021). In navigation or robotics, known forms are depth prediction (Das et al., 2018b;a; Mirowski et al., 2017), contrastive self-supervised learning (SSL) (Majumdar et al., 2022b) or privileged information from the simulator like object categories (Pashevich et al., 2021), goal directions (Marza et al., 2022), exploration (Ye et al., 2021) or visual correspondences in visuomotor policy (Florence et al., 2020; Sax et al., 2019; Hong et al., 2023). Supervised learning requires in-domain data collection that makes extension beyond the training environment and task difficult. Alternatives come from SSL (Wang et al., 2022; Xie et al., 2022). Recent pre-trained visual encoders, like DINO (Caron et al., 2021) and masked autoencoders (MAE) (He et al., 2022), have been used in (Yadav et al., 2022) and (Yadav et al., 2023), respectively. Once pre-trained, the encoder is often frozen before passing into a policy learning module. To be effective across a range of real-world robotic tasks, Radosavovic et al. (2022) diversified the image sources when pre-training. Mezghani et al. (2022) favor nearby frames to have similar visual representations.

**Pose** — *Relative Pose Estimation* (RPE) has been intensively studied in CV (Kendall et al., 2015; Kim & Ko, 2022; Xu et al., 2022). It evolved from feature matching and correspondences (Mur-Artal et al., 2015; Tang et al., 2023) to end-to-end training (Li et al., 2021; Mousavian et al., 2017) or transfer from large-scale classification (Melekhov et al., 2017) and finetuning after pre-training on geometric tasks (Weinzaepfel et al., 2022). Existing solutions revise various components of the regression pipeline (Jin et al., 2021), discretize the distribution over poses (Chen et al., 2021), etc.

Conventional RPE was developed for rather small camera displacements and assumes high visual overlap between images. *Wide-baseline RPE* refers to a challenging scenario of large view-point changes and occlusions, on which not all conventional methods work well (Jin et al., 2021). Even more challenging, navigation requires what we call *Extremely wide-baseline RPE and Visibility*, as an agent may be located in different or cluttered places and therefore may have small or no visual overlap at all. It also requires detecting overlap / visibility, which we perform in this work.

## 3 LEARNING PERCEPTION FOR GOAL ORIENTED VISUAL NAVIGATION

We target image-goal navigation in 3D environments (*ImageNav* and *Instance-ImageNav*), where an agent is asked to navigate from a starting location to a visual goal (Figure 1). The agent receives at each timestep $t$ a single image observation $\mathbf{x}_t \in \mathbb{R}^{3 \times H \times W}$ and a goal image $\mathbf{x}^* \in \mathbb{R}^{3 \times H \times W}$, both of size $112 \times 112$. The agent can select one action from the action set $\mathcal{A} = \{$MOVE FORWARD 0.25m, TURN LEFT 10°, TURN RIGHT 10°, and STOP$\}$. Navigation is considered successful if the STOP action is selected when the agent is within 1m of the goal position in terms of geodesic distance. For *Instance-ImageNav*, two additional actions are LOOK UP and LOOK DOWN.

Our objective is to learn a perception module which predicts a latent representation given an observation and goal. We conjecture that this requires the following three perception skills:

**S1 — Low-level geometric perception** of the 3D structure of the scene, which includes the detection of navigable space, obstacles, walls and exits, key elements for planning.

**S2 — Perception of semantic categories** is not only required in tasks where object categories are given as goals, which is not the case in the more challenging tasks we target, but is also an additional powerful intermediate cue for other required skills, like geometric perception. Detecting navigable space, for instance, is highly correlated with categories like *Floor, Wall*.

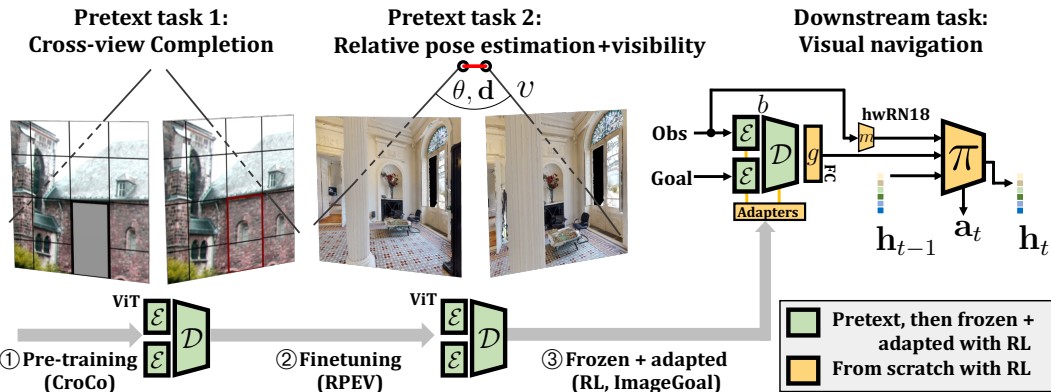

Figure 3: We address correspondence problems in image goal navigation, which we address through two pretext tasks: ① cross-view completion (Weinzaepfel et al., 2022), which reconstructs a masked image from a reference image, and ② relative pose and visibility estimation. They are learned by a binocular ViT $b$ and combined with a monocular encoder $m$ taking only observations, forming "DEBiT". Outputs are provided to a recurrent policy, maintaining memory $\mathbf{h}_t$ and predicting actions $\mathbf{a}_t$. Encoder $m$ and policy are trained with RL ③, the high-capacity model $b$ is frozen but adapted.

> **S3 — Specific object detection** and relative pose estimation under large viewpoint changes ("extremely wide baseline") is more difficult than the detection of known object classes and requires to solve a visual correspondence problem, potentially helped by semantic cues.

In end-to-end approaches, these skills have been traditionally learned directly from reward or with IL, potentially supported by additional tasks like monocular depth prediction (Das et al., 2018b;a; Mirowski et al., 2017), contrastive self-supervised learning (Majumdar et al., 2022b) or privileged information from the simulator. We argue for a more holistic approach and propose a dual visual encoder combined with a multi-step pre-training strategy. Dubbed "*DEBiT*" = *Dual Encoder Binocular Transformer*, it consists of a binocular model $b(\mathbf{x}_t, \mathbf{x}^*)$, which targets skill S3, goal detection and goal pose estimation, and a monocular model $m(\mathbf{x}_t)$, which targets skills S1 and S2 not related to the goal $\mathbf{x}^*$ — see Figure 1. The two encoders produce embeddings $\mathbf{e}_t^b$ and $\mathbf{e}_t^m$, respectively, which are integrated into a recurrent policy,

$$
\begin{aligned}
\mathbf{e}_t^b &= g(b(\mathbf{x}_t, \mathbf{x}^*)) && \text{// skill S3 – goal direction}\\
\mathbf{e}_t^m &= m(\mathbf{x}_t) && \text{// skills S1, S2}\\
\mathbf{h}_t &= f(\mathbf{h}_{t-1}, \mathbf{e}_t^b, \mathbf{e}_t^m, l(\mathbf{a}_{t-1})) && \text{// recurrent state update (agent mem)}\\
p(\mathbf{a}_t) &= \pi(\mathbf{h}_t), && \text{// policy}
\end{aligned}
\tag{1}
$$

where $g$ is a fully connected layer, $l$ is an embedding function, and $f$ is the update function of a GRU (Cho et al., 2014); for clarity we have omitted the equations of gating functions. The monocular encoder $m(\mathbf{x}_t)$ takes over the navigation skills not related to the goal, we therefore kept it reasonably small and propose a half-width ResNet-18 architecture (He et al., 2016), which is trained from scratch and from reward.

The binocular visual encoder $b(\mathbf{x}_t, \mathbf{x}^*)$ decomposes into a Siamese monocular encoder $\mathcal{E}$ applied to each image individually and a binocular decoder $\mathcal{D}$ combining both encoder $\mathcal{E}$ outputs, expressed as

$$
\mathbf{e}_t^b = g(b(\mathbf{x}_t, \mathbf{x}^*)) = g(\mathcal{D}(\mathcal{E}(\mathbf{x}_t), \mathcal{E}(\mathbf{x}^*))).
\tag{2}
$$

DEBiT is implemented as a ViT with self-attention layers in both $\mathcal{E}$ and $\mathcal{D}$ and with cross-attention layers in the decoder $\mathcal{D}$. $\mathcal{D}$ can naturally represent the correspondence problems between image patches through the attention distribution, as we will experimentally show in Section 4. For more details we refer to (Weinzaepfel et al., 2022) and to Appendix A.1.

Training the large-capacity binocular encoder entirely from scratch through reward in navigation is difficult. The underlying geometric correspondence problem is complex and can't be handled by the weak learning signal, in particular since the navigation policy needs to jointly learn multiple perception skills, plus some form of internal mapping as well as planning. Training perception separately through losses highly correlated to the perception skills we identified above, in particular S3, proved to be a key design choice — see Figure 3: we pre-train the binocular model $b$ with the *CroCo* pretext task requiring reasoning on low-level geometry (Section 3.1) and then finetune it on a novel pretext task dedicated to *ImageNav* (Section 3.2).

## 3.1 Cross-view completion

Recently, Weinzaepfel et al. (2022) introduced *Cross-View Completion* (CroCo), a potent pre-training task trained from a large amount of heterogeneous data and which captures the ability to perceive low-level geometric cues highly relevant to vision downstream tasks. It is an extension of masked image modeling (He et al., 2022) processing pairs of images $(\mathbf{x}, \mathbf{x}')$, which correspond to two different views of the same scene with important overlap. The images are split into sets of non-overlapping patches $\mathbf{p} = \{p_i\}_{i=1...N}$, and $\mathbf{p}' = \{p_i'\}_{i=1...N}$, respectively. The first input image $\mathbf{x}$ is partially masked, and the set of non-masked patches is denoted $\tilde{\mathbf{p}}$. The pretext task requires the reconstruction of the masked content $\mathbf{p} \backslash \tilde{\mathbf{p}}$ from the visible content in the second image, therefore during pre-training we replace the final FC layer $g$ by a patch-wise reconstruction layer denoted $r$,

$$\hat{\mathbf{p}} = r(b(\tilde{\mathbf{p}}, \mathbf{p}')) = r(\mathcal{D}(\mathcal{E}(\mathbf{p}), \mathcal{E}(\mathbf{p}'))). \tag{3}$$

Training minimizes the MSE loss:

$$\mathcal{L}(\mathbf{x}, \mathbf{x}') = \frac{1}{|\mathbf{p} \backslash \tilde{\mathbf{p}}|} \sum_{\mathbf{p}_i \in \mathbf{p} \backslash \tilde{\mathbf{p}}} \|\hat{\mathbf{p}}_i - \mathbf{p}_i\|^2. \tag{4}$$

CroCo is applicable to monocular and binocular downstream problems, competitive performance was shown for monocular depth estimation, optical flow and RPE (Weinzaepfel et al., 2022).

**Model and Data** — We use the publicly available[1] code from Weinzaepfel et al. (2022). We re-trained the model ourselves and also explored smaller, more robot-friendly variants. Pre-training data consists of 1.8 million image pairs rendered with the Habitat simulator.

## 3.2 Relative pose and visibility estimation in navigation

Once the binocular encoder $b$ is pre-trained with CroCo, we finetune it on a second pretext task, relative pose estimation and visibility (RPEV) for navigation settings. While for navigation purposes only a 2D vector $\mathbf{t}$ is relevant, which encodes the direction and distance from the agent to the goal, we train the prediction of the full classical *relative pose estimation* (RPE) problem, which also includes a $3 \times 3$ matrix $\mathbf{R}$ representing the relative rotation of the camera capturing the goal image w.r.t. the current agent orientation. While not useful for navigation, it can potentially add useful learning signals.

**Visibility** — Classically, accurate pose estimation assumes that two images (observation and target) share a sufficiently large part of the visual content, with the overlap providing cues sufficient to estimate the translation and rotation components from one image to the other. This assumption was also satisfied in (Weinzaepfel et al., 2022), but it is, by far, not a valid assumption in navigation. The agent is initially placed far from the goal location and is required to explore the scene, in which case the RPE task cannot be solved through geometry and correspondence, as no scene points are shared between the two images. Recent work has shown that regularities in scene layouts can be exploited to predict distributions over unseen object positions with some success (Ramakrishnan et al., 2022), but this has been reported for object categories and it is unsure whether similar results can be achieved for image exemplars.

For this reason, we added a *visibility* measure to our training data, which addresses two issues: (i) it ensures feasibility of RPE and excludes image pairs with insufficient correspondence from training the translation and rotation pose components. We do, however, train the pose components even for low amounts of overlap and treat low visibility as an extreme case ("extremely wide baseline"); (ii) it provides an additional feature to the agent, as visibility is a strong prior in both positive and negative cases. High visibility indicates closeness to the goal, which can be exploited directly through directional information $\mathbf{t}$ provided by the same model, captured in the embedding $\mathbf{e}_t^b$. Low visibility suggests to explore the scene and rather move away from the current position.

Compared to alternatives like frustrum overlap (Balntas et al., 2018), we define visibility $v \in [0, 1]$ as the proportion of patches $\mathbf{p}_i'$ of the goal image $\mathbf{x}'$ which are visible in the observed image $\mathbf{x}$. Note, that this definition is not symmetric, and exchanging the two images alters the visibility value.

**The RPEV model** — we predict the two RPE components, translation $\mathbf{t} \in \mathbb{R}^3$ and rotation matrix $\mathbf{R} \in \mathbb{R}^{3 \times 3}$, as well as visibility $v$, from an additional head $h$ (which is actually composed of three

---

[1] https://github.com/naver/croco

individual heads) attached to the binocular encoder, taking the embedding $\mathbf{e}^b$ as input,

$$(\mathbf{t}, \mathbf{R}, v) = h(\mathbf{e}^b) = h(b(\mathbf{x}, \mathbf{x}^*)), \tag{5}$$

where $\mathbf{x}$ is the observed image and $\mathbf{x}^*$ is the goal image. To ensure that $\mathbf{R}$ is a valid rotation matrix, we use orthogonal Procrustes normalization from the Roma library (Brégier, 2021).

After CroCo pre-training, we finetune the model with the following loss:

$$\mathcal{L}_{RPEV} = \sum_i \Big[ |v_i - v_i^*| + \mathbf{1}_{v_i^* > \tau} \big\{ |\mathbf{t}_i - \mathbf{t}_i^*| + |\mathbf{R}_i - \mathbf{R}_i^*| \big\} \Big], \tag{6}$$

where $i$ indexes image pairs, $\mathbf{t}_i^*, \mathbf{R}_i^*, v_i^*$ denote ground truth values, $\mathbf{1}$ is the binary indicator function, $|.|$ denotes the $L_1$ loss and $\tau$ is a threshold which switches off RPE supervision in the case of insufficient visibility.

**Dataset** — We collect a dataset tailored to perception in *ImageNav* by sampling random views from scenes in the Gibson (Xia et al., 2018), MP3D (Chang et al., 2018) and HM3D (Ramakrishnan et al., 2021) datasets. We respect the standard train/val scenes split of each dataset. We sample two points uniformly on the navigable area and query the simulator for the shortest path from one to the other. To balance the difficulty of the dataset, we split this path into 5 parts corresponding to increasing thresholds on geodesic distance ("in reach" $\leq 1$m, "very close" $\leq 1.5$m, "close" $\leq 2$m, "approaching" $\leq 4$m, and "far" $> 4$m), sample 10 intermediate positions and orientations along the path in each part, from which images are captured. We compute the fractions of pixels from the goal image visible from any of the ones captured along the path using depth frames which are then discarded. This process is repeated until 100 trajectories per scene are sampled, yielding a total of near 68.8M image pairs with position, orientation and visibility labels, representing 140GB of data.

**Training for navigation** — We train the parameters of the recurrent policy $(f, \pi)$ and the monocular encoder $m$ jointly from scratch with PPO (Schulman et al., 2017) with a reward definition in the lines of the one proposed by Chattopadhyay et al. (2021) for *PointGoal*, $r_t = K \cdot \mathbf{1}_{\text{success}} - \Delta_t^{\text{Geo}} - \lambda$, where $K = 10$, $\Delta_t^{\text{Geo}}$ is the increase in geodesic distance to the goal, and slack cost $\lambda = 0.01$ encourages efficiency. The binocular encoder is trained in two different variants:

- **Frozen** — we freeze the parameters of the binocular encoder $b$ after the two step pre-training phases (CroCo + RPEV), and then only finetune the FC layer $g$ in equation (1). Faster to train, we will use this configuration for most ablations and analyses in the experimental section.
- **Adapted** — we freeze $b$ as above, but add adapter layers as in AdaptFormers (Chen et al., 2022b), which are trained with RL jointly with the policy $(f, \pi)$ and $m$. In the next section we will show that this leads to significant performance improvements.

## 4 EXPERIMENTAL RESULTS

**Experimental setup** — We evaluate on both *ImageNav*, where the goal is a random view taken by the camera of the agent, and the more recent *Instance-ImageNav* (Krantz et al., 2022), where the goal depicts a specific object viewed from a different camera. The major parts of the experiments, ablations and analyses are performed on *ImageNav* in the classical setting, as in (Majumdar et al., 2022b; Mezghani et al., 2022). Unless stated otherwise, we trained the models for 200M steps on an A100 GPU. As in prior work, for *ImageNav* we report performance on the 14 Gibson-val scenes and thus use it as a test set, using the unseen episodes provided by Mezghani et al. (2022). For *Instance-ImageNav* we follow the protocol in (Krantz et al., 2023).

**Metrics** — RPE is evaluated over the pairs with visibility over $\tau$ in the percentage of correct poses for given thresholds on distance and angle, e.g. 1 meter and 10°. Visibility is evaluated over all pairs by its accuracy at $\pm 0.05$, i.e., the percentage of prediction within a 0.05 margin of the ground-truth value. Navigation performance is evaluated by success rate (SR), i.e., fraction of episodes terminated within a distance of $<1$m to the goal by the agent calling the STOP action, and SPL (Anderson et al., 2018), i.e., SR weighted by the optimality of the path, $SPL = \frac{1}{N} \sum_{i=1}^{N} S_i \frac{\ell_i^*}{\max(\ell_i, \ell_i^*)}$, where $S_i$ be a binary success indicator in episode $i$, $\ell_i$ is the agent path length and $\ell_i^*$ the GT path length.

**Baselines** — we compare with the state-of-the-art methods on this task, including several variants of **Siamese Encoders**, which encode the $\mathbf{x}_t$ and $\mathbf{x}^*$ separately, used by a trained policy, typically a recurrent one. First introduced by Zhu et al. (2017), they were updated by including augmented memory (Mezghani et al., 2022) and powerful ViT based architectures and self-supervised pre-training

Table 1: **Image-Nav: impact of model capacity** of the binocular encoder on RPEV and nav. perf. (CroCo+RPEV, 200M steps of RL, frozen, no adapters). $L$=layers, $H$=heads, $d$=embedd.dim.

| Variant | Encoder | | | Decoder | | | #params | Monoc | % correct poses | | Vis-acc | Nav. perf. | |
|---|---|---|---|---|---|---|---|---|---|---|---|---|---|
| | L | H | d | L | H | d | (binoc) | | 1m&10° | 2m&20° | (%) | SR (%) | SPL (%) |
| DEBiT-L ("Large"), no adapters | 12 | 12 | 768 | 8 | 16 | 512 | 120M | hwRN18 | **97.5** | **98.9** | **94.0** | 82.0 | **59.6** |
| DEBiT-B ("Base"), no adapters | 12 | 6 | 384 | 8 | 16 | 512 | 55M | hwRN18 | 92.5 | 96.8 | 89.3 | **83.0** | 55.6 |
| DEBiT-S ("Small"), no adapters | 12 | 6 | 384 | 2 | 8 | 256 | 24M | hwRN18 | 82.7 | 93.5 | 81.6 | 79.6 | 52.1 |
| DEBiT-T ("Tiny"), no adapters | 8 | 6 | 384 | 2 | 8 | 256 | 17M | hwRN18 | 80.3 | 92.4 | 80.6 | 79.3 | 50.0 |

Table 2: **ImageNav: impact of pre-training strategies**: we ablate CroCo and RPEV pre-training. All results on 100M steps of RL only, frozen, no adapters. **Left:** training performance curves (SR) for DEBiT-B, best viewed in color. **Right:** ablated test results.

| Variant | Pre-train | | % corr. poses | | Vis-acc | Nav. perf. | |
|---|---|---|---|---|---|---|---|
| | CroCo | RPEV | 1m&10° | 2m&20° | (%) | SR | SPL |
| DEBiT-L, no adapters | ✗ | ✗ | n/a | n/a | n/a | 7.0 | 4.4 |
| DEBiT-L, no adapters | ✓ | ✗ | n/a | n/a | n/a | 60.2 | 33.1 |
| DEBiT-L, no adapters | ✗ | ✓ | 40.1 | 66.7 | 58.3 | 11.8 | 9.9 |
| DEBiT-L, no adapters | ✓ | ✓ | **97.5** | **98.9** | **94.0** | **82.0** | **54.8** |
| DEBiT-B, no adapters | ✗ | ✗ | n/a | n/a | n/a | 6.8 | 4.0 |
| DEBiT-B, no adapters | ✓ | ✗ | n/a | n/a | n/a | 65.7 | 37.3 |
| DEBiT-B, no adapters | ✗ | ✓ | 39.7 | 66.4 | 58.8 | 23.6 | 17.4 |
| DEBiT-B, no adapters | ✓ | ✓ | **92.5** | **96.8** | **89.3** | **81.2** | **53.0** |

(Majumdar et al., 2022b; 2023; Yadav et al., 2023). We also compare to the feature-matching based method presented in (Krantz et al., 2023), which holds the current SOTA on *Instance-ImageNav*.

**Impact of model capacity** — we explore variations in model capacity distributed over the encoder $\mathcal{E}$ and the decoder $\mathcal{D}$ of the binocular visual encoder $b$ (the monocular part $m$ is unchanged) and introduce four different model sizes in Table 1: DEBiT-L ("*Large*"), DEBiT-B ("*Base*"), DEBiT-S ("*Small*") and DEBiT-T ("*Tiny*"), where DEBiT-L corresponds to the architecture in (Weinzaepfel et al., 2022). Performance generally improves with more model capacity.

**Impact of pre-training strategies** — Table 2 gives results comparing different pre-training strategies for the two largest variants, DEBiT-L and DEBIT-B. Directly training the binocular encoder $b$ from scratch did not lead to exploitable results, reward as a learning signal is too weak. CroCo pre-training is essential, directly training on RPEV led to low performance. CroCo pre-training alone is not optimal, RPEV adds a significant boost to the gain provided by self-supervised objective alone. The curves in Table 2 (left) shows the evolution of navigation performance (SR) during training, indicating the significant gain and head start the two pretext tasks provide.

**Aligning architecture design choices with learning signals** — visual encoders for end-to-end trained solutions in the literature for *ImageNav* are typically based on Siamese networks, where the inputs $x_t$ and $x^*$ are encoded separately, the respective embeddings are passed to current policies. This late fusion approach allows to train the models from weak reward signals, as the individual encoders learn high-level representations which are compared later in the pipeline. We claim that image comparisons of higher quality can be obtained through early fusion, where images are compared close to input on patch-level. We argue that this leads to a finer visual perception, where correspondence information is encoded in the representation in a more direct way, and provides a more useful signal to the policy. Our experiments shown in Table 3 corroborate this claim: we compare with a widely used Siamese architecture based on half-width ResNet-18 visual encoders taken from (Zhu et al., 2017) and reused in (Mezghani et al., 2022). DEBiT outperforms them when pre-trained with both pretext tasks, as CroCo pre-training allows correspondence on patch level to emerge (see further below), which leads to accurate pose estimates. Training DEBiT from reward alone is difficult. On the other hand, adding RPEV pre-training to the Siamese architecture is not helpful, the architecture based on late embedding-level fusion cannot exploit this signal.

In an additional experiment we verified whether this difference is explained by the presence of a cross-attention layer easing the computation of correspondences. We designed a hybrid architecture, dubbed (c) in Table 3, which combines convolutional Siamese encoders, implemented as a shared hwResNet18, with a Tiny cross-attention (CA) module with 2 layers, 4 heads and 256 dimensions. Performance is lukewarm, it did not manage to capture the cues provided by the pretext tasks.

Table 3: **ImageNav: aligning architecture design choices with learning signals**: when both are trained from scratch on navigation reward alone, the Siamese visual encoder (Mezghani et al., 2022; Zhu et al., 2017) performs better than our DEBiT architecture. However, DEBiT shines with self-supervised pre-training and fine-tuning, and learning signals which enable learning the correspondence problem solved by the encoder-decoder structure of the binocular stream. RPEV pre-trained models have been added a monocular encoder. Frozen, no adapters.

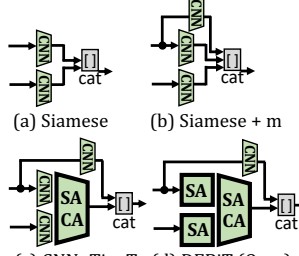

(a) Siamese   (b) Siamese + m

(c) CNN+TinyT   (d) DEBiT (Ours)

| Visual encoder | Pre-train | #parms | SR | SPL |
|---|---|---|---|---|
| (a) Siamese hwRN18* | No | 4.1M | 10.1 | 9.6 |
| (b) Siamese hwRN18*+$m$ | RPEV | 8.3M | 8.0 | 7.7 |
| (c) hwRN18+Cross-Att+$m$ | No | 10M | 7.4 | 4.7 |
| (c) hwRN18+Cross-Att+$m$ | RPEV | 10M | 7.4 | 7.2 |
| (d) DEBiT-B (Ours), no adapters | No | 60M | 6.8 | 4.0 |
| (d) DEBiT-B (Ours), no adapters | CroCo+RPEV | 60M | **83.0** | **55.6** |

*\* Baseline in (Mezghani et al., 2022), inspired by (Zhu et al., 2017)*

Table 4: **ImageNav: comparisons with prior work**: we gain +12p in SR, +12p in SPL by using RL-trained adapters of the DEBiT encoder. ANS models+weights are from (Chaplot et al., 2020b).

| Method | #steps | SR(%) | SPL(%) | Pretrained weights |
|---|---|---|---|---|
| Siam. hwRN18 | 180M | 10.1 | 9.6 | None, from scratch |
| Siam. hwRN18 [2] | 500M | - | 8.0[1] | None, from scratch |
| Mem. Aug. (Mezghani et al., 2022)[3] | 500M | - | 9.0[1] | Finetuned |
| ZSEL (Al-Halah et al., 2022) | 500M | 29.2[1] | 21.6[1] | Obs.&policy frozen, goal from scratch |
| ZSON (Majumdar et al., 2022a) | 500M | 36.9[1] | 28.0[1] | Obs. finetuned, goal frozen (CLIP) |
| VC1-ViT-L (Majumdar et al., 2023) | 500M | 81.6[1] | - | Finetuned |
| OVRL (Yadav et al., 2022) | 500M | 54.2[1] | 27.0[1] | Finetuned |
| OVRL-v2 (Yadav et al., 2023) | 500M | 82.0[1] | 58.7[1] | Finetuned |
| ANS (Chaplot et al., 2020b) + DEBiT-L | | 32.0 | 15.0 | Modular architecture + our frozen encoder |
| Ours (DEBiT-B), no adapters | 200M | 83.0 | 55.6 | Frozen |
| Ours (DEBiT-L), no adapters | 200M | 82.0 | 59.6 | Frozen |
| Ours (DEBiT-L) + adapters | 200M | **94.0** | **71.7** | Frozen + adapted |

[1] *Perf. from orig. papers;* [2] *Mono-view ablation of baseline in Table III of (Mezghani et al., 2022);* [3] *Retrained in mono-view settings, see Table 1 of (Al-Halah et al., 2022)*

**ImageNav, comparison with prior work** — Table 4 compares the proposed model with prior work. DEBiT largely outperforms the competing methods, including the memory augmented model (Mezghani et al., 2022), but also models on large-capacity ViTs like the "*Visual Cortex*" model VC1 (Majumdar et al., 2023) and OVRL2 (Yadav et al., 2023). Both have been pre-trained with masked image encoding, but in a monocular frame-by-frame basis and perform late fusion of observation and goal features, which we argue does not ease learning geometric comparisons.

**Adapters** — adding adapters to DEBIT gains additional 12p of success rate and 12p of SPL, as can be seen in Table 4. For *ImageNav*, it is unlikely that this is explained by improvement of the pose estimation performance through RL finetuning. We conjecture, that the adapters allow to pass richer information through the embedding $\mathbf{e}_t^b$ from the DEBiT to the policy.

**The *Instance-ImageNav* task** — In Table 5 we compare with the state-of-the-art in the *Instance-ImageNav* task, where the goal can be taken with arbitrary camera intrinsics (in particular FOV) and from any camera height, not necessarily the height it is installed on the agent. We trained the agent for a total of 200M steps, 100M of which were done one the *ImageNav* task followed by

Table 5: **Instance-ImageNav: adapters** enable specifying goal images with different camera intrinsics and heights compared to the obs. Performance reported on val, max/avg over the last 5 checkpoints.

| Method | #steps | — SR (%) — | | — SPL (%) — | |
|---|---|---|---|---|---|
| | | max | avg | max | avg |
| (Krantz et al., 2022) | 3500M | 5.5 | n/a | 2.3 | n/a |
| (Krantz et al., 2023) | n/a | 56.1 | n/a | 23.3 | n/a |
| Ours(DEBiT-L)+adapters | 200M | **61.1** | **59.3** | **33.5** | **32.4** |

100M on *Instance-ImageNav*. As CroCo and RPEV pre-training have been done in *ImageNav* settings (equal intrinsics), adapting DEBiT to this OOD situation was a key design choice, and with-

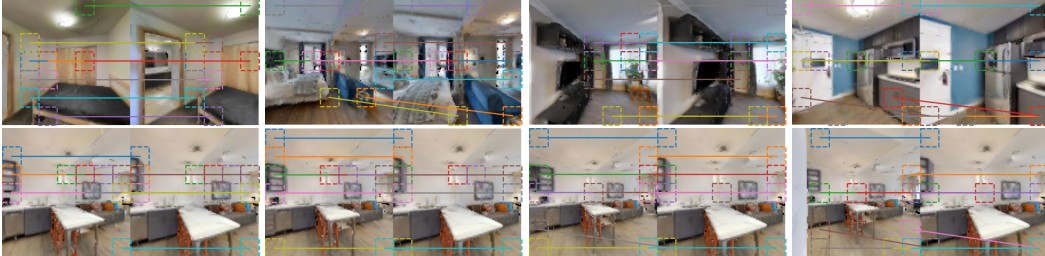

Figure 4: **Emergence of correspondence from pre-training**: visualization of the decoder cross-attention of the finetuned DEBiT-L model for example image pairs. We show attention of the last layer averaged over heads. **Top:** different scenes and poses. **Bottom:** image pairs taken from a single trajectory, varying distance to the goal, showcasing robustness to scale changes.

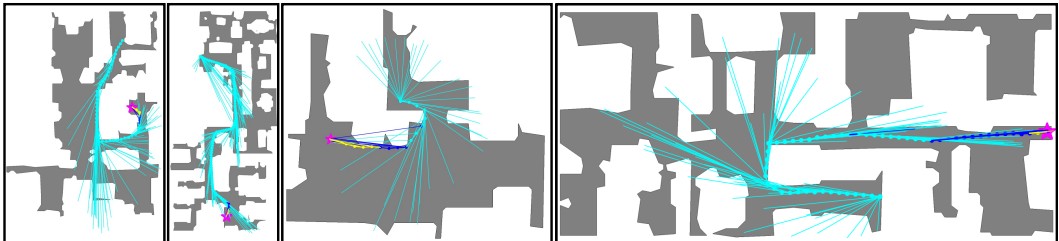

Figure 5: **RPEV performance**, DEBiT-L during expert episodes: pink star = goal; segments point to the predicted goal, color encodes visibility prediction $v$ and GT $v^*$: true negatives (TN) ($v^*<\tau, v<\tau$), TP ($v^*>\tau, v>\tau$), FP ($v^*<\tau, v>\tau$), FN ($v^*>\tau, v<\tau$) — rare, not seen. Pose and visibility (and thresholds!) are **used for visualization only**, the policy receives a latent embedding.

out adapters performance was actually unexploitable. Non-Siamese adapters (different for obs and goal) gained around 1p of SR compared to Siamese ones. We outperform the current SOTA method (Krantz et al., 2023) and show that this task can also be addressed without feature matching.

**Integration into a modular architecture** — as a proof of concept, we integrate DEBiT into the modular *exploration* method *ANS* (Chaplot et al., 2020b), which is composed of a high-level policy predicting waypoints, and a low-level navigation policy. We adapted it to *ImageNav* by adding the encoder $b$ as perception module switching between (1) nav. towards the predicted goal with the local policy or (2) exploration, otherwise, see Appendix A.5. Table 4 gives a non-comparable number, as we did not retrain ANS and took publicly available parameters. The lower performance is also explained by the direct usage of pose and visibility estimates, whereas the e2e trained models benefit from the richer latent embeddings from the visual encoders. This is confirmed by an ablation study where the e2e trained agent only receives pose and visibility, reaching only 20% SR in training.

**Visualization of attention** — in Figure 4 we visualize averaged attention of the last cross-attention layer of a DEBiT-L model. Correspondence solutions naturally emerge without explicit supervision of correspondence solutions. We show a variety of different pairs and poses in the top row, and a single trajectory varying goal distances in the bottom row, indicating robustness to scale changes.

**Visualization of RPEV performance** — Figure 5 illustrates pose and visibility estimation performance on several expert trajectories — DEBiT reliably detects the goal and provides orientations toward it. Let's recall that this information is passed to the policy indirectly through latent embeddings, the RPEV head is discarded after pre-training.

## 5 CONCLUSION

We have introduced pretext tasks and a dual visual encoder for *ImageNav* and *Instance-ImageNav* navigation, which provide rich geometric information and we show that this makes solutions of correspondence problems emerge without explicit supervision. We integrate the method into an end-to-end trained agent, which outperforms competing methods and obtains SOTA performance on both benchmarks. We also showcase the integration into a modular navigation pipeline. Future work will use the encoder for visual odometry, extend pre-trained to pairs with different camera intrinsics and/or different backgrounds, and integrate the method into a real robotics platform.

**Reproducibility** — for the sake of reproducibility, in the case of acceptance, we will provide the source code for training and evaluation based on the public Habitat_baselines codebase and links for downloading final trained model weights (CroCo + RPEV + PPO). For training, we will provide instructions for setting up the codebase, including installing external dependencies, pre-trained models and pre-selected hyperparameter configuration. For the evaluation, the code will include evaluation metrics directly comparable to the paper's results.

**Ethics statement** — we welcome the potentially high interest for society in having autonomous agents capable of various tasks like guiding customers in shopping centers, museums, hospitals and offices, or delivering parcels.

Our introduced perception modules and pretext tasks are a new step into this general direction, as they allow to improve the navigation capabilities of autonomous agents. Of course, like most scientific work in the STEM sector, this might lead to some negative societal impacts, which our work shares with most robotics applications: military robots (used in wrong hands), surveillance etc. Our direct work itself was carried out in simulation and as such is unlikely to have produced unethical results, except the impact of large-scale training on CO2 output.

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

## A  APPENDIX

All networks have been implemented in PyTorch, below we provide details of the binocular encoder $b$, the monocular encoder $m$, and the recurrent policy composed of a dynamics mapping $f$ and the prediction head $\pi$.

### A.1  THE BINOCULAR ENCODER $b$

DEBiT's binocular encoder $b$ follows the architecture in (Weinzaepfel et al., 2022), and the "*Large*" version DEBiT-L is equivalent to (Weinzaepfel et al., 2022), corresponding to the code available at https://github.com/naver/croco: the encoder $\mathcal{E}$ is a ViT-Base model, i.e., composed of $L = 12$ self-attention blocks with $H = 12$ heads each and the embedding dimension $d = 768$. The decoder $\mathcal{D}$ is composed of $L = 8$ cross-attention blocks with $H = 16$ heads each and an embedding dimension $d = 512$. We will not detail the encoder part further, as it is very similar to a standard (monocular) ViT. Concerning the cross-attention blocks used in the decoder part, they are composed of the following layers:

- A self-attention layer is applied to the features corresponding to the current view (potentially already enriched by features from the goal view in previous blocks), with pre-norm (`LayerNorm`) and skip-connection.
- The resulting features are used as queries on the goal view features (used as keys and values, optionally also pre-normalized) in a cross-attention layer, with skip-connection.
- A 2-layers perceptron with dimensions $d = 2048$ and back to $d = 512$ independently projects the features of each patch, with pre-norm, GELU activation and skip-connection.

Full details on these cross-attention blocks are available in (Weinzaepfel et al., 2022). Smaller DEBiT versions differ in the number of layers, heads and the embedding sizes, see Table 1.

### A.2  THE PROJECTION $g$

The decoder blocks are followed by a single patch-wise linear layer $d$, which can also be seen as a $1 \times 1$ convolution on the features of all patches in 2D. It projects them from dimension $512$ to $64$, before flattening them to a 3136-dim vector (given images of size $112 \times 112$ and patches of size $16 \times 16$).

### A.3  AUXILIARY HEADS $h$ FOR RPEV

For the second phase of pre-training, the result of the flattened projection $g$ of size $3136$ is ReLU-activated and fed to a common 1024-dim linear layer, before being dispatched to $3$ independent output layers for the predictions of relative camera translation $\mathbf{t}$ and rotation $\mathbf{R}$, as well as goal visibility from current view $v$:

- The translation head directly outputs a 3D vector in the coordinate frame of the current view.
- The rotation head outputs a 9D vector which is reshaped as a $3 \times 3$ matrix, constrained to be a valid rotation matrix (using orthogonal Procrustes normalization) with a small regularization term added to the loss.
- The goal visibility head linearly outputs a single value with no activation constraining it to be between $0$ and $1$.

### A.4  THE MONOCULAR ENCODER $m$

The monocular encoder is a half-width ResNet-18 as frequently used in prior work on visual navigation (Mezghani et al., 2022; Majumdar et al., 2022b). It is very similar to a standard ResNet-18 (He et al., 2016), only differing in 3 ways:

1. Instead of using $64$, $128$, $256$ and $512$ channels in the $4$ layers (of $2$ basic blocks each), the half-width ResNet-18 uses $32$, $64$, $128$, and $256$ channels.

2. All `BatchNorm2D` layers are replaced by `GroupNorm` layers with 16 groups each.

3. The final (global pooling + linear layer) is replaced by a small "Compression" module which consist in: a 3x3 convolution (with padding) reducing the number of channels from 256 to 128, followed by a `LayerNorm` and a ReLU activation, whose result is flattened and fed to a linear layer to produce a 512-dim flat embedding of the current (monocular) view.

## A.5   THE RECURRENT POLICY $(f, \pi)$

The policy relies on a single-layer GRU as our recurrent state encoder. The 3 flat features vectors produced by the binocular, monocular, and previous action encoders are concatenated and fed to the GRU, whose output $h_t$ is passed to 2 linear heads that respectively generates a softmax distribution over the action space (Actor head), and an evaluation of current state (Critic head). This agent is structurally similar to the agent from (Ramakrishnan et al., 2022; Wijmans et al., 2019), which is used for the related point-goal navigation task, and achieves 100% PointGoal success on the Gibson dataset (Xia et al., 2018), but with different input modalities (RGB + ImageGoal instead of RGB-D + PointGoal).

## A.6   THE ANS ADAPTATION TO *ImageGoal*

Figure 6 shows our adaptation of *Active Neural SLAM* (Chaplot et al., 2020b) to the *ImageGoal* task. The new components are shown in orange. They include the target image, the binocular encoder $b$ as additional perception module, and a module switching between (1) navigation towards the predicted goal with the local policy and (2) exploration using the global+local policy, otherwise. Switching is done by thresholding the visibility prediction $v_t$ with a threshold $T$, whose influence we tested in sensibility study below.

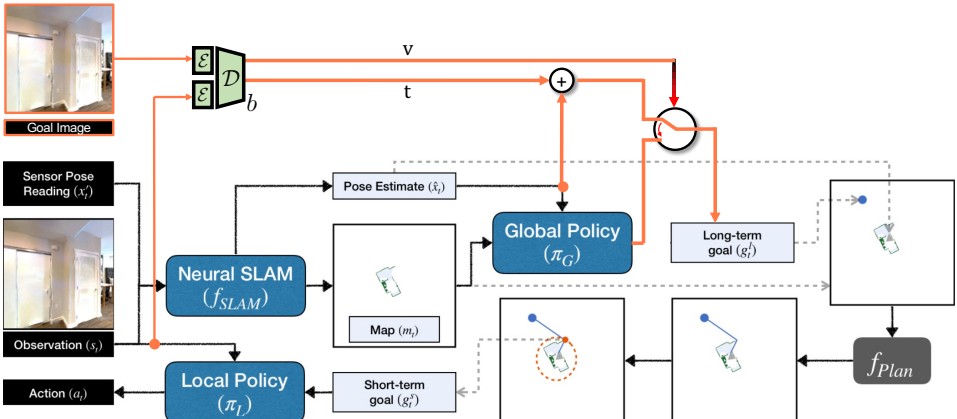

Figure 6: Active Neural SLAM + DEBiT-L/$b$ — we use the binocular encoder $b$ only of the DEBiT architecture. Figure is reproduced from (Chaplot et al., 2020b), with additional parts from our adaptation to *ImageGoal* drawn in orange.

## A.7   COMPUTATIONAL COMPLEXITY

Below we report computational complexity of the *training* pipeline in terms of processed frames per second (fps) on a single NVIDIA A100 GPU. This includes forward and backward passes as well as the overhead of running simulation in 12 parallel Habitat environments. Note, that on a real robot only forward pass over the visual encoder and policy is required.

In terms of complexity of the correspondences themselves, it is the complexity of attention, which is quadratic in terms of tokens (which are patches) and linear in the embedding dimension, number of heads and number of layers. We have input images of size $112 \times 112$ and patch size of $16 \times 16$ which gives $7 \times 7 = 49$ patches per image.

| Model | FPS |
|---|---|
| DEBiT-L | 92 |
| DEBiT-B | 156 |
| DEBIT-S | 192 |
| DEBIT-T | 225 |

Table 6: Time complexity for different DEBiT models.

## A.8 THE INSTANCE-IMAGENAV BENCHMARK

Instance-ImageNav task has been trained and tested on the Habitat-Matterport3D (HM3D) Semantics v0.2 dataset that was provided for the Habitat Navigation Challenge (Habitat, 2023). The dataset contains 216 different scenes; the train/val/test split is 145/36/35. Our model was trained on the 145 training scenes, we reported results for the 36 validation scenes. The remaining 35 test scenes are used by organizers internally; they are not publicly available.

## A.9 VISUALIZATION OF CORRESPONDENCES

The visualization in Figure 4 is simple: we have examined the last cross-attention layers, with attention summed over all heads, and looked at individual attention values. We have then picked the highest N attention values in this matrix and displayed their corresponding query-key pairs, i.e., drawing a line between the corresponding patches.

## A.10 THE HWRN18+CROSS-ATT ARCHITECTURE

The architecture of the binocular encoder for variant "hwRN18+Cross-Att" used in Table 3 of the main paper is given as follows:

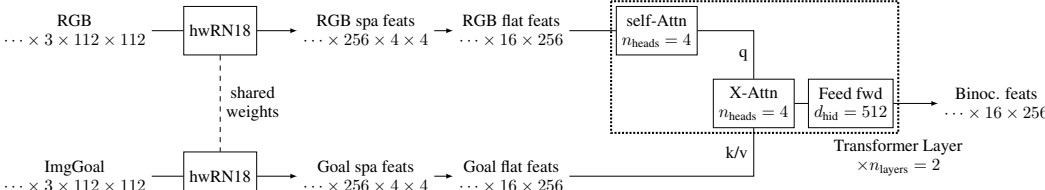

In the sake of clarity, normalization and skip-connections in the transformer decoder are not represented on the Figure. Contrary to standard ViTs, patch size is 1 (because ResNet already reduced the input image size), no positional embeddings are added to the tokens, and no class tokens are used.

## A.11 ABLATIONS AND SENSITIVITY ANALYSIS OF $\tau$ IN RPEV

**Controlling** $\tau$ — Our RPEV model is trained to predict the visibility as well as the relative pose only on image pairs with visibility above $\tau = 0.2$. We now propose ablations on the RPEV model where we remove the visibility predictions, as well as where we vary $\tau$ to 0.1 and 0.4. Results are reported in Table 7. We report the visibility accuracy (predicted visibility in the range of 0.05 from the ground-truth) as well as the percentages of correctly predicted poses within two error thresholds (1m&10°and 2m&20°) computed over pairs with different visibility overlap $\tau'$. Overall, the performance is close for all models. When a model is trained for a larger $\tau$ than used for evaluation ($\tau > \tau'$), the performance is a bit worse given that the visibility might be too low compared to what have been seen during training, explaining the 2 or 3% lower values for the model trained with $\tau = 0.4$.

Summary: $\tau$ controls the threshold during training, $\tau'$ controls the threshold in evaluation.

**Removing visibility altogether** — If a pure RPE model were trained, i.e. RPEV without visibility, the model could be pretrained only on image pairs with overlap, i.e. visibility $> 0$. Indeed, it would be impossible for a model to predict a relative pose without any relevant information when the visibility is zero. On the other hand, when extracting features with this model for navigation, many input pairs would have no overlap, i.e., visibility, setup that would not have been seen during

the pre-training. To avoid this out-of-distribution setting, we believe it is important to also pretrain on pairs without overlap during RPEV, explaining why we include it in all experiments.

To verify that predicting visibility does not harm the RPE performance, we have trained a relative pose estimation model that is trained only on pairs with visibility above the threshold $\tau$, but without any visibility prediction or supervision. These results are provided in Table 7, first line, the impact is negligible.

Table 7: **Ablations on RPEV.** We measure the impact of having the visibility and of varying $\tau$, the threshold on the visibility to decide whether to use a given pair as training and report the visibility accuracy as well as the percentages at correct poses at two thresholds, for 3 different visibility thresholds $\tau'$ of minimum visibility between pairs. The impact of these variants is overall limited.

| | Visibility Head | Vis-acc (%) | $\tau' = 0.2$ | | $\tau' = 0.3$ | | $\tau' = 0.4$ | |
|---|---|---|---|---|---|---|---|---|
| | | | 1m&10° | 2m&20° | 1m&10° | 2m&20° | 1m&10° | 2m&20° |
| RPE ($\tau = 0.2$) | ✗ | - | 92.4 | 96.8 | 93.5 | 97.4 | 93.5 | 97.5 |
| RPEV ($\tau = 0.2$) | ✓ | 89.3 | 92.4 | 96.8 | **93.6** | **97.4** | **93.7** | **97.6** |
| RPEV ($\tau = 0.1$) | ✓ | **90.0** | **92.5** | **96.9** | 93.3 | 97.3 | 92.9 | 97.4 |
| RPEV ($\tau = 0.4$) | ✓ | 87.7 | 89.6 | 95.6 | 92.7 | 97.1 | 93.5 | 97.5 |

## A.12 LIMITATIONS

Higher image resolutions than $112 \times 112$ could boost the goal recognition. On Instance-ImageNav, pre-training could sample image pairs with different camera intrinsics, as this is currently only learned through RL.

## A.13 VIDEO

This appendix is accompanied by an **additional video** showing and explaining rollouts of the agents.

