# OpenReview forum: "End-to-End (Instance)-Image Goal Navigation through Correspondence as an Emergent Phenomenon"
_ICLR.cc/2024/Conference — ICLR 2024 poster_

### Official Review · Reviewer_1KiF · 2023-10-30

**Soundness:** 4 excellent
**Presentation:** 3 good
**Contribution:** 3 good
**Rating:** 8
**Confidence:** 5

**Summary:**

In this work, the authors introduce pretext tasks and a dual visual encoder for ImageNav and Instance-ImageNav navigation, offering rich geometric information and enabling the resolution of the challenging mono-view scenario with end-to-end trained techniques.

The method breaks down the problem into multiple training stages, demonstrating the emergence of solutions to the correspondence problem without explicit supervision.

Through experiments, the manuscript highlights the effectiveness of the proposed pretext tasks and a dedicated dual encoder architecture, surpassing competing methods and achieving  a novel state-of-the-art performance on both benchmarks. Additionally, the authors demonstrate seamless integration into a modular navigation pipeline of the proposed approach.

**Strengths:**

- In the state of the art (Section 2), a detailed review of all the literature related to the problem investigated in this work is conducted. Furthermore, the main differences of the proposed model with respect to previous works are outlined.

- The system proposed in Section 2 is novel. It describes an integration of subsystems/submodules that allows for an approximation to the problem, yielding highly promising results. Additionally, the proposed design can be considered innovative. The integration of Cross-View Completion (CroCo) into a navigation system as described has not been explored before, although it's true that this contribution is not the strongest of all. The second pretext task, relative pose estimation and visibility (RPEV) for navigation settings, is interesting and provides significant results. Overall, the originality of the work is considerable.

- The experimental evaluation provides very interesting results in which the proposed model outperforms the state of the art (see tables 4 and 5). An ablation study is provided where the real impact of each of the contributions or parts of the model can be interpreted. Finally, qualitative evidence is presented on how the proposed pre-training process results in the emergence of correspondences between images by analyzing the attention of the last layers.

**Weaknesses:**

The experimental evaluation included in the paper has some minor limitations that should be addressed:
- It is not clear on which databases or environments the system has been tested. For example, on what environments are the results reflected in Table 5 obtained? The interested reader has to review (Krantz et al., 2023) to know these details. Section 4, "Experimental setup" subsection needs to be improved.

- I believe one of the most interesting contributions of the proposed model is that it generates correspondences between images due to how the pre-training is designed. Figure 4 shows some results or qualitative evidence in this regard. However, the paper does not explain in detail how this image is generated or how these correspondences are analyzed. The manuscript simply mentions that they "visualize averaged attention of the last cross-attention layer of a DEBiT-L model", but more details should be provided.

**Questions:**

Overall, I see an strong paper here.
I would simply suggest to the authors that they provide explanations for the limitations I have described in the previous section.

**Details Of Ethics Concerns:**

Authors already provide a convincing ethics discussion.

---

> ### Author Response · Authors · 2023-11-21
>
> We thank you for your review.
>
> > It is not clear on which databases or environments the system has been tested. For example, on what environments are the results reflected in Table 5 obtained? The interested reader has to review (Krantz et al., 2023) to know these details.
>
> Instance Imagenav task has been trained and tested on the Habitat-Matterport3D (HM3D) Semantics v0.2 2 dataset that was provided for the Habitat Navigation Challenge 2023. The dataset contains 216 different scenes; the train/val/test split is 145/36/35. Our model was trained on the 145 training scenes, we reported results for the 36 validation scenes. The remaining 35 test scenes are used by organizers internally; they are not publicly available.
>
> This information has been added to the appendix of the paper.
>
> > I believe one of the most interesting contributions of the proposed model is that it generates correspondences between images due to how the pre-training is designed. Figure 4 shows some results or qualitative evidence in this regard. However, the paper does not explain in detail how this image is generated or how these correspondences are analyzed. The manuscript simply mentions that they "visualize averaged attention of the last cross-attention layer of a DEBiT-L model", but more details should be provided.
>
> The visualization is simple:  we have examined the last cross-attention layers, with attention summed over all heads, and looked at individual attention values. We have then picked the highest N attention values in this matrix and displayed their corresponding query-key pairs, i.e., drawing a line between the corresponding patches.
>
> This has been added to the paper and to the video.

---

### Official Review · Reviewer_isxL · 2023-10-31

**Soundness:** 4 excellent
**Presentation:** 4 excellent
**Contribution:** 3 good
**Rating:** 6
**Confidence:** 4

**Summary:**

This paper improves the view correspondence encoding in image-goal navigation by pre-training a large binocular ViT model (DEBiT) with cross-view completion (CroCo) and relative pose estimation and visibility prediction (RPEV) from images in photorealistic indoor scenes. Specifically, the dual encoder in DEBiT takes binocular images as input, and the encoded features are merged with a decoder to model the correspondence between the two images. DEBiT is pre-trained with CroCo, followed by RPEV to estimate the relative distance, relative rotation, and overlap between two images. When addressing downstream Instance-ImageNav and ImageNav tasks, DEBiT takes the goal image and agent's observation as inputs, its network is adapted in tuning, and the output representations will be passed to the policy network for decision-making. Results show substantial improvement compared to previous approaches.

**Strengths:**

- This paper studies an important problem in visual navigation: learning high-capacity perception modules for modeling and reasoning view correspondence. It proposes DEBiT, which implements a dual encoder to enable early fusion of the images. It introduces CroCo and RPEV pre-training, which are highly relevant and have shown to be effective in addressing the problem.
- Significant improvement is achieved compared to previous approaches, boosting the ImageNav and Instance-ImageNav results to 94% SR and 59.3% SR, respectively. The methods introduced in this paper and the resulting pre-trained models are very likely to inspire/to be used by future research in relevant fields.
- The paper is technically sound; many important arguments and design choices are justified by experiments. Besides the key results, there are many highly constructive analyses/findings, such as "directly training binocular encoder from scratch", "tuning visual encoder with downstream policy network", "early fusion vs. late fusion", "attention with scale changes", etc., that are very valuable to future research.
- Overall, this paper is nicely written; it is very compact, informative, and clear. All methods (and most of the implementation), visualizations, and discussions are clearly presented.

**Weaknesses:**

- The proposed DEBiT model seems to be limited in addressing image-goal navigation tasks (the image-nav task itself needs more justification), and it is unclear how it might benefit other visual navigation problems.
    - However, I do believe that CroCo and RPEV can help in learning a general (and better) navigation-specific perception model (e.g., for obj-Nav, language-guided-Nav, Audio-Nav, etc.), while I am concerned that it might be much less effective compared to ImageNav which is defined to take two images as input.

- I am aware that ImageNav is an interesting visual navigation problem and has gained some research attention (publications), but I am still not convinced by its setting, especially how much practical value it might bring to real-world applications. [This question might be more appropriate for researchers who proposed ImageNav, but since this paper is devoted to ImageNav, I believe it must have a very strong reason and motivation behind.]
    - The two ImageNav tasks consider indoor short-range navigation for finding static objects (Instance-ImageNav has paths with an avg. geodesic length of 12.41m); from the user's perspective (e.g., a household robot), giving instruction in the form of a specific image is unnatural.
    - From the ImageNav data statistics and the visualization provided in this paper (supp. video), it seems that most of the paths have very few intersections, and the agent doesn't really need to explore the environment but keeps moving to new regions until the target is in its sight. This could be the reason why a very simple policy network, without methods like SLAM, can still lead to amazingly high performance (94% SR). I am concerned that the ImageNav task itself oversimplifies the problem, and this paper might overclaim the contribution to visual navigation research.

- Some experiments are missing to justify arguments (see Questions below).

- The limitation and future extension of this work are not discussed in this paper.

**Questions:**

Questions without a star (*) are not critical to my evaluation of this paper, but I still hope the authors can kindly and briefly respond to them. Please also respond to my concerns in Weaknesses.

- (*) This paper mentioned depth inputs, but I wonder why depth images are not applied in the model since it might greatly facilitate learning view correspondence and identifying space and obstacles. (I might have overlooked some details; please correct me if I did.)

- (*) How does the choice of $\tau$ influence the pre-training and the downstream results? If two images are too distant away and have little overlap, will it be too noisy to learn? Or it might help the visual encoder to learn distant image correspondence and benefit exploration in navigation? Any numerical analysis on this?

- (*) The paper claims the benefit of having visibility estimation but does not quantify its impact on downstream tasks. Any results for this?

- Just curious, can CroCo and RPEV be trained simultaneously?

- In Table 2, from Tiny to Large, the model size increases drastically, but the difference in navigation results is quite small; what might be the reason? Does it mean that a very large perception model might not be necessary? What would be the result of DEBiT-Tiny + Adapters?

- Instance-ImageNav depicts targets viewed from a different camera; if this is the main reason why the proposed method gets much less improvement compared to ImageNav (an OOD situation as claimed), I wonder what if the images for pre-training are augmented with different camera parameters, (1) Is it still feasible to learn CroCo and RPEV? (2) Will it reduce the visual domain gap in downstream?

- Are the visual encoders in OVRLs fine-tuned with the policy networks in downstream tasks? From their papers and Table 4, it seems Yes. But in the section Related Work, "Once pre-trained, the encoder is often frozen before passing into a policy learning module."

- About comparison experiments.
    - Using adapters in the perception model largely improves the results. I wonder how much improvement it might bring to the previous approaches.
    - In Table 3, for a fair comparison (in terms of pre-training tasks and #params), I think DEBiT-Tiny and RPEV-only should be listed, which I believe is more rigorous and can prove the same argument.
    - (*) In Table 4, an important point to mention is the size of the visual encoders (#params) applied in each work. After scanning some papers in the Table, I believe there is a clear trend of larger-models-better-results. But it seems that DEBiT is relatively efficient, especially compared to OVRL-v2. I hope the authors can clarify this point, which I believe will also strengthen the argument.

Other Suggestions:
- Remove "(Instance)-" from the title.
- I found a paper, "Learning navigational visual representations with semantic map supervision (Hong et al., ICCV2023)", which also focuses on learning better navigation visual encoder with view correspondence and uses two images for pre-training. It seems relevant and could be added to the references.

---

> ### Author Response · Authors · 2023-11-21
>
> We thank you for your review.
>
> > The proposed DEBiT model seems to be limited in addressing image-goal navigation tasks
>
> We plan to use DEBiT also for visual odometry, in the lines of:
>
> Partsey et al. Is mapping necessary for realistic pointgoal navigation? CVPR 2022.
>
> > ImageNav (...) [This question might be more appropriate for researchers who proposed ImageNav, but since this paper is devoted to ImageNav, I believe it must have a very strong reason and motivation behind.]
>
> We indeed cannot answer for the original authors introducing the two benchmarks, but we do agree on the observations. However, this task was among the most difficult among the EAI navigation tasks until recently, much more than ObjectNav for instance, and we think we have contributed to making significant steps forward. We actually also do agree on your point which, if we have understood correctly, states that visual search can be separated from classical observation tasks like exploration and detecting (and respecting) navigable space. This is the exact reason why we proposed a specialized architecture with tailored pre-text losses, which addresses exactly this issue. We argue that this will still hold if the length of the episodes increases significantly, as this will require more exploration skills, not more competence in visual comparison.
>
> We do not claim that DEBiT will contribute to the exploitation of visual priors as other work does, but which would be complementary: if, for instance, our goal image shows a shampoo bottle which DEBiT does not find in the current observation, the exploration component could exploit the semantic information in the goal and guide the robot towards bathrooms.
>
> > The limitation and future extension of this work are not discussed in this paper.
>
> Limitations:
> * further improvements could be done on the Instance-ImageNav benchmark by changing the pre-training tasks such that image pairs are also sampled with different camera intrinsics, as currently this difference is only learned through the RL loss and with adapters.
> * increasing the image resolution might be beneficial.
>
> Future and ongoing work will address the following aspects:
> * using DEBiT for visual odometry,
> * Pre-training on frame pairs which correspond to the Instance-ImageNav setting
> * Training the model on pairs where the background of the goal image is segmented out by a model like “Segment Anything”
> * Porting the model to our navigation software on our real robotics platform
>
> This has been added to the paper.
>
> > (*) This paper mentioned depth inputs, but I wonder why depth images are not applied in the model
>
> We did not use depth for multiple reasons:
> * we do not want to rely on capturing depth for the goal image, as this would significantly restrict the use case of ImageNav.
> * we think that depth is extremely helpful in navigation but perhaps less so when calculating visual correspondences. The boost it could probably get in simulation might quickly get offset when the model is run on a real robot, as the sim2real gap is high.
> * the two benchmarks we also targeted for evaluation do not use depth images.
>
> > (*) How does the choice of tau influence the pre-training and the downstream results?
>
> We have added an ablation by training the RPEV with tau=0.1 and tau=0.4, compared to tau=0.2 in the paper. We report the results in the table below and added it to the appendix of the revised manuscript (Table 7). More precisely, we report the visibility accuracy (predicted visibility within 0.05 of the ground-truth one) as well as the percentages of correct poses within 1m&10° and 2m&20°, considering all pairs with visibility above a threshold tau’, for tau’ = 0.2, 0.3 and 0.4.
>
> Overall, in terms of RPEV performance, the impact is extremely limited. We are training the navigation model with these new visual encoder variants, but this will not be ready during the rebuttal phase. We will add them to the camera ready version of the paper.
>
> | tau        |   vis-acc | % correct poses tau’=0.2 | % correct poses tau’=0.3 | % correct poses tau’=0.4 |
> |-----------|-------------|-----------------------------------|-----------------------------------|----------------------------------------|
> **0.2** | 89.3 | 92.4 / 96.8 | **93.6** / **97.4** |    **93.7** / **97.6**  |
> 0.1       | **90.0** | **92.5** / **96.9** |  93.3 / 97.3  |   92.9 / 97.4 |
> 0.4       |  87.7 | 89.6 / 95.6 | 92.7 / 97.1 |    93.5 / 97.5 |
>
> This table has been added to the appendix.

---

> > ### Author Response · Authors · 2023-11-21
> >
> > > (*) The paper claims the benefit of having visibility estimation but does not quantify its impact on downstream tasks.
> >
> > We have trained a RPE model without visibility (with tau=0.2), results below, as well as in the revised appendix (Table 7). We observe that the performance is similar. However, the binocular decoder of a model for RPE only (without visibility) would have trained only on pairs with visibility above tau, while in navigation, many pairs will have a zero visibility. We are now training the navigation on top of it and expect lower performance for this reason. As said above, this will not be ready during the rebuttal phase.
> >
> > | tau=0.2        |   vis-acc | % correct poses (tau’=0.2) |
> > |-----------|-------------|-----------------------------------|
> > | RPEV | 89.3 | 92.4 / 96.8 |
> > | RPE   | -      | 92.4 / 96.8 |
> >
> > This is part of Table 7 in the revised appendix.
> >
> > > Just curious, can CroCo and RPEV be trained simultaneously?
> >
> > The order of pre-text tasks has been chosen to optimize the compromise between the amount of information in the learning signal and its correlation to the downstream task:
> > * CroCo provides a dense learning signal, since reconstruction is supervised densely, e.g. for each patch. This process allows the high-capacity model to implicitly solve the correspondence problem necessary for most binocular downstream tasks.
> > * RPEV provides a global (e.g. not dense) learning signal and as such provides less information as CroCo, but this information is highly relevant to the ImageNav downstream task. It also requires the solution of the correspondence problem to be solved, and we argue that it makes sense to do this in a subsequent step.
> >
> > In this, we follow the strategy already proposed in the original CroCo paper, which sequentially trained for CroCo first and then finetuned for a variety of downstream tasks. This is still the case for the CroCo follow-up work (Weinzaepfel et al., CroCo v2: Improved Cross-view Completion Pre-training for Stereo Matching and Optical Flow, ICCV 2023), and which seems to be first-ranked on the stereo-vision benchmark https://spring-benchmark.org/stereo.
> >
> > Training the two tasks simultaneously is perhaps worth exploring but it was not our priority for the moment.
> >
> > > In Table 2, from Tiny to Large, the model size increases drastically, but the difference in navigation results is quite small
> >
> > The difference from T to L is +9.6p in SPL and +2.7p in SR (w/o adapters), we think that this is quite a significant gain. Running DEBiT-L on the real robot should not be an issue if the robot has an onboard GPU. Our current platform (we did not port DEBiT-L to it yet) supports decision delays of up to 300ms while allowing fast real time navigation with 1m/s. This is well within the range of DEBiT-L.
> >
> > > Instance-ImageNav (...) I wonder what if the images for pre-training are augmented with different camera parameters
> >
> > This is certainly possible and planned. The main work load would be to export new frame pairs from the Instance-ImageNav benchmark and we think that further gains could be achieved. This will be left for future work.
> >
> > > Are the visual encoders in OVRLs fine-tuned with the policy networks in downstream tasks?
> >
> > OVRL is indeed finetuned, and probably an exception to the trend.
> >
> > > Using adapters in the perception model largely improves the results. I wonder how much improvement it might bring to the previous approaches.
> >
> > OVRL (for instance) has been finetuned, we do not think that adaptation will bring additional gains. As we pointed out in the section “Aligning architecture design choices with learning signals”, the situation is different for late-fused type models like OVRL, which can largely benefit from fine-tuning, and these models have very likely already reached their optimal adaptation. On the other hand, the early fusion at patch-level done by DEBiT is potentially much more powerful, as we argue, but requires a stronger supervision signal. Finetuning with RL was not successful in our experiments, and this is the reason why the adapters were needed and did help.
> >
> > > In Table 3 (...) I think DEBiT-Tiny and RPEV-only should be listed
> >
> > Sorry for the confusion, the term “Tiny” in lines (x) of table does not refer to DEBiT-Tiny. It was a term we chose to indicate a small two-layer cross-attention module.
> >
> > We renamed it to “hwRN18+Cross-Att” and added the architecture details to the appendix.
> >
> > > (*) It seems that DEBiT is relatively efficient
> >
> > We argue that the main efficiency comes from the binocular nature of the encoder and the cross-attention layers, together with CroCo pre-training. One can only go so far by pre-training on still images for a problem which requires solving an underlying image correspondence problem.
> >
> > > Remove "(Instance)-" from the title.
> >
> > We will check with the AC whether this can be done.
> >
> > > I found a paper (Hong et al., ICCV2023)"
> >
> > Thank you for the reference, it has been added to the paper.

---

> ### Comment · Reviewer_isxL · 2023-11-22
> **Final Rating**
>
> I'm satisfied with most of the responses and appreciate the additional analyses and discussions from the authors. After reading all the other reviews and responses, I decided to keep my rating as Weak Accept (6).

---

### Official Review · Reviewer_Sgek · 2023-10-31

**Soundness:** 3 good
**Presentation:** 4 excellent
**Contribution:** 3 good
**Rating:** 6
**Confidence:** 5

**Summary:**

This paper is concerned with image-goal navigation and proposes a new end-to-end method for this task. The main contribution of the work is the inclusion of a pre-training stage of two auxiliary tasks that help in learning relevant visual features to be used as a state representation to a recurrent policy. The method achieves state-of-the-art performance on publicy available datasets.

**Strengths:**

The paper is well written and easy to follow. The authors show that they understand the underlying challenges of this problem well and clearly explain their thought process behind the proposed approach.

I agree with the authors that the bottleneck for these navigation problems is perception. I think the pretext tasks being proposed help in learning relevant representations for this task which would be otherwise very difficult to learn in typical end-to-end methods. Overall I think the paper proposes  a novel and sensible approach and moves the needle forward on this subject and improves the sample efficiency of these methods.

**Weaknesses:**

The title is somewhat misleading as the emergence of correspondences is really not the focus of the work, but more of an afterthought. The content of the paper might be mistaken as an investigation into this phenomenon that is carried out in this paper:
[A] Tang et al, Emergent of Correspondences from Image Diffusion, arXiv, 2023.
In fact the emergence of correspondences from the learned representation is not really that surprising, given that the pre-training task is relative pose estimation. Even before the introduction of transformers, monocular pose estimation methods have shown that the representation learns to identify meaningful keypoints on objects. One example:
[B] Mousavian et al, 3D Bounding box estimation using deep learning and geometry, CVPR 2017

I appreciate the inclusion of the experiment where DEBiT was integrated with ANS for a direct comparison to a modular approach. However, I am surprised by the large performance gap to the proposed method. It seems unintuitive that a method that tries to map pixels directly to actions would outperform an approach that uses a map and de-couples the planning from the control. I think this should be looked more carefully to ensure fair comparison. Was the global policy of ANS finetuned with the frozen binocular encoder b and using the AdaptFormers? ANS was trained for object-goal so probably the Neural SLAM and global policy components need to be re-trained for the ImageGoal task. As it stands, the experiment is not convincing that end-to-end methods are better than modular approaches.

**Questions:**

Why is using panoramas treated as unrealistic with regards to robotic applications? RGB sensors (and even RGB-D) are relatively cheap and can be easily mounted on robots to cover a 360 degree field-of-view.

The authors cover some of the literature on pre-text tasks for improving sample efficienty but I think these are also worth discussing:
[C] Ye et al, Auxiliary Tasks and Exploration Enable ObjectGoal Navigation, ICCV 2021
[D] Sax et al, Mid-Level Visual Representations Improve Generalization and Sample Efficiency for Learning Visuomotor Policies, CoRL 2019
Especially regarding [D], shouldn't a representation trained on a multitude of vision tasks be more task-generalizable to just pre-training on relative pose estimation?

---

> ### Author Response · Authors · 2023-11-21
>
> We thank you for your review.
>
> >The title is somewhat misleading as the emergence of correspondences is really not the focus of the work, but more of an afterthought. The content of the paper might be mistaken as an investigation into this phenomenon that is carried out in this paper: [A] Tang et al, Emergent of Correspondences from Image Diffusion, arXiv, 2023. In fact the emergence of correspondences from the learned representation is not really that surprising, given that the pre-training task is relative pose estimation. Even before the introduction of transformers, monocular pose estimation methods have shown that the representation learns to identify meaningful keypoints on objects. One example: [B] Mousavian et al, 3D Bounding box estimation using deep learning and geometry, CVPR 2017
>
> While the emergence of correspondence is not the focus of this work, we nevertheless think that it is an interesting finding, in particular when compared to competing methods for this problem, which either calculate correspondences explicitly (Krantz et al, 2023), or are based on late fusion, where correspondences is hard or impossible to emerge. We therefore argue that this sets our work apart from the literature on this topic.
>
> Thank you for the references, we have added them to the paper.
>
> > (...) ANS (...). However, I am surprised by the large performance gap to the proposed method. It seems unintuitive that a method that tries to map pixels directly to actions would outperform an approach that uses a map and de-couples the planning from the control. I think this should be looked more carefully to ensure fair comparison. Was the global policy of ANS finetuned with the frozen binocular encoder b and using the AdaptFormers? ANS was trained for object-goal so probably the Neural SLAM and global policy components need to be re-trained for the ImageGoal task. As it stands, the experiment is not convincing that end-to-end methods are better than modular approaches.
>
> ANS was not finetuned, we took the weights from the model provided by the authors. This will also explain a drop in performance. As said in the paper, this experiment was proof-of-concept of the possible integration of DEBiT into a modular method. It should not be seen as a direct comparison. We have toned down the language on the comparison in the paper.
>
> We conjecture that a large gain is obtained by the fact that the end-to-end trained policy benefits from the richer latent embeddings passed from the pen-ultimate layers of the visual encoders, whereas ANS only receives pose and visibility directly. We have run an experiment to confirm this where  the end-to-end trained agent only receives pose and visibility. It reached only 20\% SR in training, which provides evidence that the full embedding is quite important for the end-to-end trained agent. We conjecture that the performance could be further increased by longer training, but we made it comparable and trained the agent for 200M steps.
>
> We have added some elements of this to the paper.
>
> > Why is using panoramas treated as unrealistic with regards to robotic applications? RGB sensors (and even RGB-D) are relatively cheap and can be easily mounted on robots to cover a 360 degree field-of-view.
>
> In the panoramic setting, both observed and goal images are supposed to be panoramic, and this is why the setting is not realistic. While 4 cameras can be simply put on a robot, the panoramic setting also requires that 4 different *goal* images are captured by user, which is often done with a hand-held camera or a cell phone, and it is required to do this with the same optical center and with exact angles of 90°. This is very complicated to do for an end user.
>
> > The authors cover some of the literature on pre-text tasks for improving sample efficienty but I think these are also worth discussing: [C] Ye et al,, ICCV 2021 [D] Sax et al,, CoRL 2019 Especially regarding [D], shouldn't a representation trained on a multitude of vision tasks be more task-generalizable to just pre-training on relative pose estimation?
>
> Thank you for these references, we will add them to the paper. We indeed think that certain / several pre-text tasks are complementary to ours, and it would also certainly be worth exploring whether these tasks should be used to train the monocular encoder m, or whether the binocular encoder should take over parts of the reasoning they add.

---

> > ### Comment · Reviewer_Sgek · 2023-12-02
> >
> > The authors have answered my comments at a satisfactory level and I believe they generally did a good job in their rebuttal. After reading the other reviews and the authors' responses I am keeping my recommendation of acceptance.

---

### Official Review · Reviewer_Q8jf · 2023-11-06

**Soundness:** 3 good
**Presentation:** 3 good
**Contribution:** 3 good
**Rating:** 8
**Confidence:** 4

**Summary:**

The authors introduced pretext tasks and a dual visual encoder for ImageNav and Instance-ImageNav navigation in 3D environments, which provide rich geometric information and make it possible to address the challenging mono-view setting with end-to-end trained methods. Via experiments it is shown that peformance on competing methods and SOTA on both benchmarks is better.  The idea of cross-view completion and goal direction computation as pre-text for ImageGoal in contrast to ObjectGoal is the philosophical novelty in the work. Apart from that, the pipeline and architecture presented will aid future scope of research in the benchmark challenges.

**Strengths:**

Fig. 3 is well described to give the view of the problem scope being breaked into cross view completion, relative pose estimation and visual navigation.
This is a well written paper with clear explainations and technical soundess.
The results sections and ablation studies uphold the claims.
This paper will help the ImageGoal community - hence recommended.

**Weaknesses:**

The last portion of supplementary video in terms of correspondence needs better representation and also that is the core focus.
The analyis of time complexity for doing correspondances in the benchmark and the distribution of work load should have helped understand the bottlenecks for a near real time system like robotic agents, even in embodied setups.
A practical deployment in robotic setup should have confirmed the real world transfer applicability.
I think Fig. 1 image you ahve search to related the chair with big picture - can any other image or better reoslution be used?
Same with panaromic and mono view - please help in making sense of the image if at all included in body. If space contsraint, appendix referral is there for later sections, but introduction setup has to be clear.
I think related work can only focus on the core related work, getting rid of object goal and general visual nav in such detail - this space can be used elsewhere in explaination later.
No limitations of the work is presented. Future gaps should be explained well.

**Questions:**

"we split this path into 5 parts cor" - any logic regarding the discreet steps evenly spaced?
Instead of Active Neural SLAM, anything else has been tried out in the pipeline?

---

> ### Author Response · Authors · 2023-11-21
>
> We thank you for your review.
>
> > The last portion of supplementary video in terms of correspondence needs better representation and also that is the core focus.
>
> We have added an explanation to the video.
>
> In essence, the visualization is simple: we have examined the last cross-attention layers, with attention summed over all heads, and looked at individual attention values. We have then picked the highest N attention values in this matrix and displayed their corresponding query-key pairs, i.e., drawing a line between the corresponding patches.
>
> >The analysis of time complexity for doing correspondances in the benchmark and the distribution of work load should have helped understand the bottlenecks for a near real time system like robotic agents, even in embodied setups.
>
> Computational complexity is dominated by the visual encoder, and our training frame-rates indeed dropped when adding the attention based binocular encoder. However, we still get high frames rates in training on an A100:
>
> * DEBiT-L : 92 fps
> * DEBiT-B : 156 fps
> * DEBiT-S : 192 fps
> * DEBiT-T : 225 fps
>
> This includes all forward passes over visual encoders and policy, and also things not needed in inference on a robot: backward passes and the simulation of 12 different Habitat environments in parallel. We did not yet test DEBiT on our own robots, but we have tested similar policies running on an embedded Nvidia Jetson and we can achieve decisions with delays of well under 100ms, which allows the robot to move quickly with 1m/s if dynamics if the delay is handled correctly in simulation as well.
>
> In terms of complexity of the correspondences themselves, it is the complexity of attention, which is quadratic in terms of tokens (which are patches) and linear in the embedding dimension, number of heads and number of layers. We have input images of size 112x112  and patch sizes of 16x16 which gives 7x7 = 49 patches per image.
>
> This has been added to the appendix of the revised paper.
>
> > A practical deployment in robotic setup should have confirmed the real world transfer applicability.
>
> We leave this for future work, integration in our real robotics platform will actually start very soon.
>
> > I think Fig. 1 image you ahve search to related the chair with big picture - can any other image or better reoslution be used? Same with panaromic and mono view - please help in making sense of the image if at all included in body.
>
> We have zoomed into the observation in the revised paper, this should make the chair better visible.

---

### Author Response · Authors · 2023-11-21
**Common answer**

We thank the reviewers for the effort they put into reading, studying and evaluating our work. We are happy that they appreciated
- a **well written paper with clear explanation**,
- a **strong paper** which **outperforms the state of the art** and provides **significant improvement**,
- that **this paper moves the needle forward on this subject**, and **will help the ImageGoal community**, **the system** … **is novel**.
- **technical soundness**,
- that **the authors show that they understand the underlying challenges**,
- that **the results sections and ablation studies uphold the claims**, **many important arguments and design choices are justified by experiments**.

We provided answers to the reviewers’ questions individually, as usual.

We also updated the paper, the appendix, all changes have been made visible by orange color. We also updated the video (one more title frame with explanations, as requested)

---

### Meta-Review · Area_Chair_KvM4 · 2023-12-10

**Metareview:**

The submission studies the image-goal navigation problem with two pretext tasks: cross-view completion and relative pose estimation. Reviewers like the simple solution, strong results, and interesting discovery of the emergence of correspondence. All reviewers recommended acceptance, and the AC agreed. Reviewers still pointed out some issues on how generally applicable the method is and if the title is appropriate. The authors are encouraged to further address these concerns during the camera ready version.

**Justification For Why Not Higher Score:**

Reviewers have concerns regarding the general applicability of the paper.

**Justification For Why Not Lower Score:**

All reviewers are supportive of acceptance.

---

### Decision · Program_Chairs · 2024-01-16

Accept (poster)